# SVQN: Sequential Variational Soft Q-Learning Networks

**Shiyu Huang, Hang Su, Jun Zhu**[*]**, Ting Chen**[*]
Dept. of Comp. Sci. & Tech., BNRist Center, Institute for AI, THBI Lab, Tsinghua University
hsy17@mails.tsinghua.edu.cn; {suhangss, dcszj, tingchen}@tsinghua.edu.cn

## Abstract

Partially Observable Markov Decision Processes (POMDPs) are popular and flexible models for real-world decision-making applications that demand the information from past observations to make optimal decisions. Standard reinforcement learning algorithms for solving Markov Decision Processes (MDP) tasks are not applicable, as they cannot infer the unobserved states. In this paper, we propose a novel algorithm for POMDPs, named sequential variational soft Q-learning networks (SVQNs), which formalizes the inference of hidden states and maximum entropy reinforcement learning (MERL) under a unified graphical model and optimizes the two modules jointly. We further design a deep recurrent neural network to reduce the computational complexity of the algorithm. Experimental results show that SVQNs can utilize past information to help decision making for efficient inference, and outperforms other baselines on several challenging tasks. Our ablation study shows that SVQNs have the generalization ability over time and are robust to the disturbance of the observation.

## 1 Introduction

In recent years, substantial progress has been made in deep reinforcement learning for solving various challenging tasks, including the computer Go game (Silver et al., 2016), Atari games (Mnih et al., 2015), StarCraft (Zambaldi et al., 2018; Pang et al., 2018) and the first-person shooting (FPS) games (Lample & Chaplot, 2017; Wu & Tian, 2016; Huang et al., 2019). However, in many real-world applications, decision-making problems are partially observable (Aström, 1965), preventing such problems from being solved by standard reinforcement learning algorithms. Formally, these kinds of problems are often defined as Partially Observable Markov Decision Processes (POMDPs) (Kaelbling et al., 1998), which demand information from past observations to help in the decision-making process (McCallum, 1993).

Although numerous efforts (Hausknecht & Stone, 2015; Foerster et al., 2016; Igl et al., 2018; Zhu et al., 2018) have been paid to tackle this problem, there still exist various challenges. For example, Egorov (2015) tries to solve POMDPs by using the belief of the agent as the input of DQN (Mnih et al., 2015), but this algorithm needs access to the environment model. However, in many reinforcement learning tasks, it is not possible for the agent to acquire the underlying transition function, making such algorithms inapplicable. Some recent work (Karkus et al., 2017; McAllester & Singh, 2013; Babayan et al., 2018) tries to solve POMDPs under the model-free setting, i.e., the agent does not need to know and learn the transition function of the environment. For instance, Karkus et al. (2017) trained an agent to navigate in a partially observable grid world under the model-free setting, i.e., the agent can only observe a part of the grid world and does not learn the transition function. The agent uses its local observations to update its beliefs (McAllester & Singh, 2013; Babayan et al., 2018). In their experiments, the ground truth of the state is the full map plus the location of the agent, which means the representation of the state is explicit. However, in some complex tasks, it is impossible to acquire or design the state or beliefs.

To solve the unknown representation problem of the state, Hausknecht & Stone (2015) and Zhu et al. (2018) try to represent the state as latent variables of neural networks. However, they only use

---

[*]J. Zhu and T. Chen are corresponding authors. J. Zhu is also with RealAI.

a deep recurrent neural network to capture the historical information and fail to utilize the Markov property of the state in POMDPs. Igl et al. (2018) apply sequential Monte Carlo (SMC) (Le et al., 2017) to introduce inductive bias to the neural network, which can embody the Markov properties of the state. They can infer the state from the past observations online. However, they separate the planning algorithm from the inference of the state.

To infer the hidden states and optimize the planning module jointly, we represent POMDPs as a unified probabilistic graphical model (PGM) and derive a single evidence lower bound (ELBO). We apply structured variational inference to optimize the ELBO. In our implementation, we design generative models to infer the hidden variables, however, the distribution of the latent variables is conditioned on previous hidden states. This is different from standard VAEs (Kingma & Welling, 2013), whose prior of the latent variables can be a standard Gaussian distribution. So, we apply an additional approximate function to tackle the conditional prior problem. The planning can also be solved under the PGM framework. Fortunately, maximum entropy reinforcement learning (MERL) (Levine, 2018) provide a tool to formalize the planning as a probabilistic inference task.

In this paper, we propose a novel end-to-end neural network called the sequential variational soft Q-learning network (SVQN), which integrates the learning of hidden states and the optimization of the planning within the same framework. A deep recurrent neural network (RNN) (Cho et al., 2014; Hochreiter & Schmidhuber, 1997) in SVQNs is used to reduce the computational complexity, because the feature extraction can share the same weights in the RNN. Experimental results show that the SVQN can utilize past information to help in decision making for efficient inference, and outperforms other baselines on several challenging tasks. Our ablation study shows that SVQNs have the generalization ability over time and are robust to the disturbance of the observation.

**Contributions:** (1) We derive the variational lower bound for POMDPs, which allows us to integrate the optimization of the control problem and learning of the hidden state under a unified graphical model. (2) We propose to tackle the difficulty of the inference of the hidden state and solve the problem of a conditional prior using the generative models. (3) We design an end-to-end deep recurrent neural network, which can reduce the computational complexity and be trained efficiently.

## 2 RELATED WORK

We summarize some related work in POMDPs and inference methods for sequential data.

**Model-Based and Model-Free methods for POMDPs:** When the environment model is accessible, POMDPs can be solved by model-based method. Egorov (2015) used model-based methods to solve POMDPs, but their agents need to know the belief-update function and the transition function. When the environment model is unknown, model-free methods should be applied. Recently, some researchers (Hausknecht & Stone, 2015; Zhu et al., 2018) used recurrent neural networks to capture the historical information, but they failed to utilize the Markov property of the state in POMDPs. Our work proposes generative models for the algorithm learning, which tackles the difficulty of the inference of hidden states and introduces inductive bias to the network structure. Igl et al. (2018) applied sequential Monte Carlo (SMC) to the POMDPs. They can infer the hidden state from the past observations online. However, they separate the planning algorithm and the inference of the hidden state. Our algorithm is derived from a unified graphical model, which can train the inference model and the planning algorithm jointly.

**Explorations in POMDPs:** In contrast to MDP methods, POMDP methods should do exploration to gather information with no immediate reward which can then be used in the future to gain higher rewards. Pathak et al. (2017) and Choi et al. (2018) designed deep reinforcement learning algorithms to help the exploration in their tasks. Actually, their tasks are POMDP tasks. However, they just treat their tasks as MDP tasks and intuitively add exploration tricks to standard reinforcement learning algorithms. Their exploration tricks include the reconstruction of observations and actions, which comes from intuitive concepts such as curiosity and attention. The generative models in our method also need to reconstruct the observations and actions, but our algorithm is derived from solid theoretical foundations.

**Inference for Sequential Data:** The observation in POMDP tasks is sequential data, and we need to do inference for the hidden state from observations. Coquelin et al. (2009) used a particle filter to estimate the belief state given past observations. However, their method needs to access to the

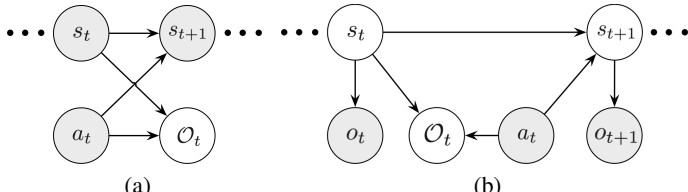

(a)  (b)

Figure 1: The graphical models for Markov decision processes (MDPs) (a) and partially observable Markov processes (POMDPs) (b). Grey nodes are observed, white nodes are hidden. In POMDPs, the state $s$ is not observable and must be inferred from past observations. The $\mathcal{O}_t$ is a binary random variable, where $\mathcal{O}_t = 1$ denotes that the action is optimal at time $t$, and $\mathcal{O}_t = 0$ denotes that the action is not optimal.

environment model. Chung et al. (2015) derived the evidence lower bound of latent variables for sequential data and designed a novel deep neural network to infer recurrent latent variables. Our inference method has some superficial resemblances to their method, but we derived the evidence lower bound from a different graphical model and tackle the difficulty of how to integrate the RL planning algorithm instead of just inferring the hidden variables.

## 3 PRELIMINARIES

We start by briefly reviewing the backgrounds and notations related to our method, including partially observable Markov decision processes and maximum entropy reinforcement learning which solves optimal control problems using a probabilistic framework.

### 3.1 PARTIALLY OBSERVABLE MARKOV DECISION PROCESSES

In many control tasks, the complete state of the environment is unknown to the agent. Instead, the agent can only receive local observations, which are typically conditioned on the current state of the system. The agent needs to make decisions based on the historical information. This kind of problems can be formalized as partially observable Markov decision processes (POMDPs) (Smith & Simmons, 2004). Formally, a POMDP is represented as a tuple $(S, A, O, T, Z, r)$, where $S$, $A$ and $O$ are state space, action space and observation space, respectively. The reward function $r(s, a)$ is the received reward when taking action $a$ in state $s$. $T(s, a, s')$ is the state-transition function, which defines the probability of the succeeding state $s'$ after taking action $a$ in state $s$. $Z(s, a, o)$ is the observation function, which defines the probability of emitted observation $o$ after taking action $a$ in state $s$.

POMDPs use the belief $b_t(s)$ to maintain the distribution of the unknown state at time $t$, and the agent updates its belief with a Bayesian filter when receiving a new observation as

$$b_t(s') = \eta Z(s', a_t, o_t) \sum_{s \in S} T(s, a_t, s') b_{t-1}(s), \tag{1}$$

where $\eta$ is a normalizing constant and $a_t$ is the action at time $t$. POMDP planning needs to find a policy $\pi$ to maximize the accumulative reward as

$$V_\pi(b_0) = \mathbb{E}(\sum_{t=0}^{T} \gamma^t r(s_t, a_t) | b_0, \pi), \tag{2}$$

where $s_t$ is the state at time $t$, and $\gamma \in (0, 1)$ is a discount factor.

### 3.2 MAXIMUM ENTROPY REINFORCEMENT LEARNING

POMDPs require solving two problems: the inference of state and optimal control. To integrate both of them under a unified framework, we represent POMDPs as a probabilistic graphical model. So the optimal control problem needs to be solved as a probabilistic inference task. Fortunately, the maximum entropy reinforcement learning (MERL) algorithm provides an effective tool to solve optimal control problem under PGM framework. The graphical model of Markov decision processes

is shown in Fig. 1(a). We borrow the notation from Levine (2018) to illustrate the algorithm. Levine (2018) introduces a binary random variable $\mathcal{O}_t$ to the graphical model, where $\mathcal{O}_t = 1$ denotes that the action is optimal at time $t$, and $\mathcal{O}_t = 0$ denotes that the action is not optimal. The probability distribution of $\mathcal{O}$ is $p(\mathcal{O}_t = 1|s_t, a_t) = \exp(r(s_t, a_t))$, and the variational lower bound is given by:

$$\log p(\mathcal{O}_{1:T}) \geq \mathbb{E}_{(s_{1:T}, a_{1:T}) \sim \pi(s_{1:T}, a_{1:T})} \left[ \sum_{t=1}^{T} r(s_t, a_t) - \log \pi(a_t|s_t) \right], \tag{3}$$

where $\pi(a|s)$ is the policy function. Standard reinforcement learning only needs to maximize the cumulative reward. However, MERL will maximize an extra term, which is the policy entropy at each visited state. To get the optimal solution for MERL, two messages are introduced, i.e., $\beta_t(s_t, a_t) = p(\mathcal{O}_{t:T}|s_t, a_t)$ and $\beta_t(s_t) = p(\mathcal{O}_{t:T}|s_t)$, and their relations are given by:

$$\beta_t(s_t) = \int_A \beta_t(s_t, a_t) p(a_t|s_t) da_t,$$
$$\beta_t(s_t, a_t) = \int_S \beta_{t+1}(s_{t+1}) p(s_{t+1}|s_t, a_t) p(\mathcal{O}_t|s_t, a_t) ds_{t+1}, \tag{4}$$

and the optimal policy is given by:

$$\pi(a_t|s_t, \mathcal{O}_{t:T}) = \frac{\beta_t(s_t, a_t)}{\beta_t(s_t)}. \tag{5}$$

We can define the Q-value function and V-value function as below:

$$Q(s_t, a_t) = \log \beta_t(s_t, a_t), \tag{6}$$
$$V(s_t) = \log \beta_t(s_t), \tag{7}$$

and the update functions are:

$$V(s_t) = \log \int_A p(a_t|s_t) \exp(Q(s_t, a_t)) da_t,$$
$$Q(s_t, a_t) = r(s_t, a_t) + \log \mathbb{E}_{s_{t+1} \sim p(s_{t+1}|s_t, a_t)}[\exp(V(s_{t+1}))]. \tag{8}$$

To maximize the variational lower bound in Eq. (3), we can use a parameterized Q-function $Q_\theta(s_t, a_t)$, where $\theta$ is the function parameter, and it can be learned via gradient descent:

$$\theta = \theta - \alpha \mathbb{E}[\frac{Q_\theta(s_t, a_t)}{d\theta} - (r(s_t, a_t) + \log \int_A p(a_{t+1}|s_{t+1}) \exp(Q_\theta(s_{t+1}, a_{t+1})) da_{t+1})], \tag{9}$$

where $\alpha$ is the learning rate. This algorithm is called soft Q-learning, because the update function for Q-value can be considered a soft-update version of the standard Bellman backup. For a better understanding of MERL, we recommend readers reference this tutorial (Levine, 2018).

## 4 SEQUENTIAL VARIATIONAL SOFT Q-LEARNING NETWORKS

We now present our algorithm in detail. We first derive the variational lower bound for POMDPs, and then illustrate how to deal with the conditional prior.

### 4.1 VARIATIONAL LOWER BOUND FOR POMDPS

Different from MDPs, the state $s$ of POMDPs in the PGM (shown in Fig. 1(b)) is unobservable, which need to be inferred from the action $a$ and the observation $o$. We need to derive different variational lower bound for POMDPs, which can be used to infer the hidden state and do planning jointly.

Unlike the observation function $Z(s, a, o)$ in Section 2.1, we assume that the observation is emitted from the hidden state $s$, which means the probability distribution of observations are only conditioned on states, i.e., $o_t \sim p(o_t|s_t)$. This assumption can hold in many tasks. For example, the observation in a partially observed maze is only determined by the full map and the agent's location.

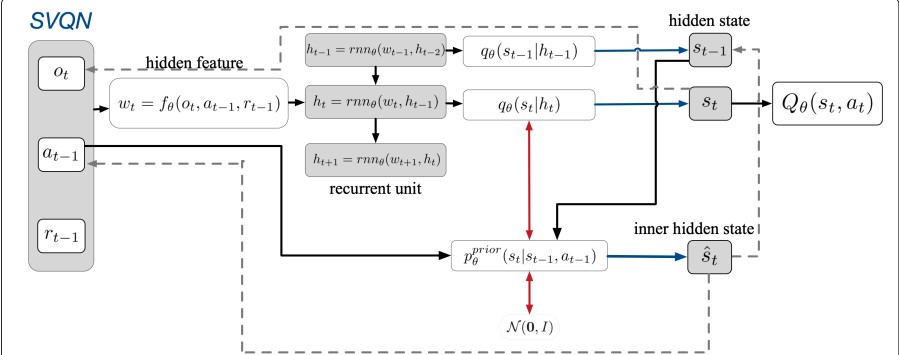

Figure 2: The structure of sequential variational soft Q-learning networks (SVQNs). Black solid lines represent forward paths of the neural network, gray dashed lines represent reconstruction paths and blue arrows stand for sampling latent variables using the re-parameterization trick. Double-arrows indicate that the algorithm needs to minimize the KL-divergence between two probability distributions. The model takes the observation $o_t$, previous action $a_{t-1}$ and reward $r_{t-1}$ as inputs, and it uses a neural network $f_\theta(o_t, a_{t-1}, r_{t-1})$ to extract the low-dim hidden feature $w_t$. The recurrent unit $rnn_\theta(w_t, h_{t-1})$ is used to capture the historical information $h_t$. $q_\theta(s_t|h_t)$ and $p_\theta(s_t|s_{t-1}, a_{t-1})$ are proposed distributions of hidden states. The hidden state $s_t$ and inner hidden state $\hat{s}_t$ are sampled from $q_\theta(s_t|h_t)$ and $p_\theta^{prior}(s_t|s_{t-1}, a_{t-1})$ respectively. The Q-function $Q_\theta(s_t, a_t)$ is learned via the temporal difference (TD) algorithm with soft update.

We apply the structured variational inference to optimize the evidence lower bound of POMDPs. In structured variational inference, different parts of the proposal distributions can be optimized separately, which means we can fix some approximate functions and optimize other approximate functions. In POMDPs, we will use two approximate functions $q_\pi(a_t|s_t)$ and $q_\theta(s_t|s_{t-1}, a_{t-1}, o_t)$. $q_\pi(\cdot)$ approximates the optimal policy and $q_\theta(\cdot)$ approximates the function of inferring hidden states, where $\theta$ is the parameter of the approximate function. When $q_\theta(\cdot)$ is fixed, the learning procedure is same as MERL, so that $q_\pi(\cdot)$ can be learned via the soft Q-learning algorithm. Conversely, when $q_\pi(\cdot)$ is fixed as the optimal policy, we can learn the inference function $q_\theta(\cdot)$ for hidden states. We can derive the evidence lower bound(ELBO) as bellow:

$$
\begin{aligned}
\log p(\mathcal{O}_{0:T}, a_{0:T}, o_{1:T}) &= \log \mathbb{E}_{q_\theta(s_{1:T}|\mathcal{O}_{1:T}, a_{0:T}, o_{1:T})} \left[ \frac{p(s_{1:T}, \mathcal{O}_{0:T}, a_{0:T}, o_{1:T})}{q_\theta(s_{1:T}|\mathcal{O}_{0:T}, a_{0:T}, o_{1:T})} \right] \\
&\geq \mathbb{E}_{q_\theta(s_{1:T}|\mathcal{O}_{1:T}, a_{0:T}, o_{1:T})} \log \left[ \frac{p(s_{1:T}, \mathcal{O}_{0:T}, a_{0:T}, o_{1:T})}{q_\theta(s_{1:T}|\mathcal{O}_{0:T}, a_{0:T}, o_{1:T})} \right],
\end{aligned}
\tag{10}
$$

and the ELBO can be written as:

$$
\begin{aligned}
\mathcal{L}(\mathcal{O}_{0:T}, a_{0:T}, o_{1:T}) = \mathbb{E}_{q_\theta(s_{1:T}|\mathcal{O}_{1:T}, a_{0:T}, o_{1:T})} \sum_{t=1}^{T} &\Big\{ r(s_t, a_t) + \log\left[ p(a_t)p(o_t|s_t) \right] \\
&- \mathcal{D}_{KL}\left[ q_\theta(s_t|s_{t-1}, a_{t-1}, o_t) || p(s_t|s_{t-1}, a_{t-1}) \right] \Big\}.
\end{aligned}
\tag{11}
$$

To get the optimal action in POMDPs, we need to maximize the ELBO above. The first term in the ELBO is the cumulative reward $\sum_{t=1}^{T} r(s_t, a_t)$, which can be optimized via maximum entropy reinforcement learning algorithm.

We apply generative models to optimize the rest terms in the ELBO. The term $p(o_t|s_t)$ means that the hidden state $s_t$ needs to have the ability to generate current observation $o_t$. The Kullback-Leibler divergence term $\mathcal{D}_{KL}\left[ q_\theta(s_t|s_{t-1}, a_{t-1}, o_t) || p(s_t|s_{t-1}, a_{t-1}) \right]$ indicates that we should minimize the gap between approximate function $q_\theta(s_t|s_{t-1}, a_{t-1}, o_t)$ and the prior $p(s_t|s_{t-1}, a_{t-1})$. In standard VAEs, the prior $p(\cdot)$ is generally fixed and can be set as a standard Gaussian. However, in this generative model, the prior of the hidden state $s_t$ is conditioned on previous state $s_{t-1}$ and action $a_{t-1}$. We will show how we deal with the conditional prior in next sub-section. More details about the derivation process of the ELBO can be found in Appendix A.

## 4.2 VARIATIONAL AUTOENCODERS FOR THE CONDITIONAL PRIOR

The KL-divergence term in Eq. (11) introduces a conditional prior for the hidden state $s_t$, but the true conditional distribution $p(s_t|s_{t-1}, a_{t-1})$ is unknown, making it intractable to do inference. Hence we propose a new parameterized function $p_\theta^{prior}(s_t|s_{t-1}, a_{t-1})$ to approximate the true distribution $p(s_t|s_{t-1}, a_{t-1})$. We use a standard VAE to learn the $p_\theta^{prior}(\cdot)$. Hence $(s_{t-1}, a_{t-1})$ can be treated as the inner observed data in this VAE model and the inner hidden state $\hat{s}_t$ is inferred from $(s_{t-1}, a_{t-1})$. We can write down the ELBO for the conditional prior directly by using the ELBO of standard VAEs (see in Appendix B):

$$
\begin{aligned}
\mathcal{L}(s_{t-1}, a_{t-1}) = &-\mathcal{D}_{KL}(p_\theta^{prior}(s_t|s_{t-1}, a_{t-1})||p(s_t)) \\
&+ \mathbb{E}_{p_\theta^{prior}(s_t|s_{t-1}, a_{t-1})}\left[\log p(s_{t-1}, a_{t-1}|s_t)\right],
\end{aligned}
\tag{12}
$$

where $p(s_t)$ can be set as a standard Gaussian. The $p_\theta^{prior}(\cdot)$ can be learned via a standard VAE paradigm.

The outputs of $p_\theta^{prior}(\cdot)$ are $\mu^{prior}$ and $\sigma^{prior}$, so that the KL-divergence term in Eq. (11) can be written as:

$$
\begin{aligned}
&\mathcal{D}_{KL}\left[q_\theta(s_t|s_{t-1}, a_{t-1}, o_t)||p(s_t|s_{t-1}, a_{t-1})\right] \\
&\approx \mathcal{D}_{KL}\left[q_\theta(s_t|s_{t-1}, a_{t-1}, o_t)||p_\theta^{prior}(s_t|s_{t-1}, a_{t-1})\right] \\
&= \mathcal{D}_{KL}\left[q_\theta(s_t|s_{t-1}, a_{t-1}, o_t)||\mathcal{N}(\mu^{prior}, \text{diag}(\sigma^{prior^2}))\right].
\end{aligned}
\tag{13}
$$

Unlike standard VAEs, the second distribution of the KL-divergence function in Eq. (13) is not a standard normal distribution. But we can still get the final loss function if we expand the KL-divergence function. We present the final formula of the loss function for Eq. (13) in Appendix C.

## 4.3 SEQUENTIAL VARIATIONAL SOFT Q-LEARNING NETWORKS

We design a deep recurrent neural network to improve the ELBO derived in Section 4.1 and Section 4.2. Fig. 2 shows the overall structure of our algorithm.

In our implementation, there are two generative models, i.e., one of the generate model learns the conditional prior $q_\theta^{prior}(\cdot)$ and reconstructs the inner observed data $(s_{t-1}, a_{t-1})$, and another generate model learns the $q_\theta(\cdot)$ in Eq. (13) and reconstructs the current observation $o_t$.

For the first generative model, there are two losses, i.e., $L_{KL}^{inner}$ and $L_{MSE}^{inner}$. $L_{KL}^{inner}$ is the negated KL-divergence between $p_\theta^{prior}(s_t|s_{t-1}, a_{t-1})$ and a standard Gaussian:

$$
L_{KL}^{inner} = -\mathcal{D}_{KL}[p_\theta^{prior}(s_t|s_{t-1}, a_{t-1})||\mathcal{N}(\mathbf{0}, I)].
\tag{14}
$$

And $L_{MSE}^{inner}$ is the reconstruction loss of $(s_{t-1}, a_{t-1})$:

$$
L_{MSE}^{inner} = MSE\left((s_{t-1}, a_{t-1}), \varphi_\theta^{inner}(\hat{s}_t)\right),
\tag{15}
$$

where $MSE(\cdot)$ is the mean-square error function, the inner hidden state $\hat{s}_t$ is sampled from $p_\theta^{prior}(s_t|s_{t-1}, a_{t-1})$, and $\varphi_\theta^{inner}(\cdot)$ is a reconstruction function.

For the second generative model, there are also two losses, i.e., $L_{KL}^{elbo}$ and $L_{MSE}^{elbo}$. $L_{KL}^{elbo}$ is defined as:

$$
L_{KL}^{elbo} = -\mathcal{D}_{KL}\left[q_\theta(s_t|s_{t-1}, a_{t-1}, o_t)||p_\theta^{prior}(s_t|s_{t-1}, a_{t-1})\right].
\tag{16}
$$

And $L_{MSE}^{elbo}$ is the reconstruction loss of $o_t$:

$$
L_{MSE}^{elbo} = MSE\left(o_t, \varphi_\theta^{elbo}(s_t)\right),
\tag{17}
$$

where $MSE(\cdot)$ is the mean-square error function, the hidden state $s_t$ is sampled from $q_\theta(s_t|s_{t-1}, a_{t-1}, o_t)$, and $\varphi_\theta^{elbo}(\cdot)$ is a reconstruction function.

Figure 3: Screen shots of Atari games and ViZDoom tasks. From left to right, they are Pong, ChopperCommand, DoubleDunk, Asteroids, Health Gather,Health Gather v2 and Defend Center. When only given one frame as the input, these Atari games are POMDPs because we can not obtain the velocity of the moving object from a single observation. Because the agent in ViZDoom environment can only see in just one direction, it naturally introduces partial observability to these tasks.

Finally, we get two kinds of loss functions for the two generative models, i.e., the reconstruction loss $L_{MES} = L_{MSE}^{inner} + L_{MSE}^{elbo}$ and the KL-divergence loss $L_{KL} = L_{KL}^{inner} + L_{KL}^{elbo}$.

And for the planning algorithm, we use the soft Q-learning algorithm (Levine, 2018). Its loss function is the temporal difference error $L_{TD}$, which is updated via Eq. (9). All these losses can jointly be optimized via stochastic gradient descent algorithms.

To reduce the computation complexity, we use a deep recurrent neural network to capture the historical information. We first use the $f_\theta(o_t, a_{t-1}, r_{t-1})$ to extract a low-dim hidden feature $w_t$ from current input, and then feed this features to a recurrent unit $rnn_\theta(\cdot)$ to get the recurrent output $h_t$. Because $h_t$ contains the information of past observations, the loss function in Eq. (16) can be rewritten as:

$$L_{KL}^{elbo} = -\mathcal{D}_{KL}\left[q_\theta(s_t|h_t)||p_\theta^{prior}(s_t|s_{t-1}, a_{t-1})\right]. \tag{18}$$

In the training, we tried both LSTM cell (Hochreiter & Schmidhuber, 1997) and GRU cell (Cho et al., 2014) as basic recurrent units. Because of the generalization ability of the recurrent neural network (Lample & Chaplot, 2017), we can just sample a fixed length $H$ of sequential data for training instead of using the full-length data. In the experiment, we also studied how the training length $H$ influences the final performance. We use a parallel training strategy, i.e., the program hosts multiple games in parallel and they all send batched data to a central data memory. This is quite similar to the data collection method in ELF (Tian et al., 2017). More details about the training strategy can be found in Appendix E.

## 5 EXPERIMENTS

We evaluate our algorithm on flickering Atari (Hausknecht & Stone, 2015) and ViZDoom platform (Kempka et al., 2016). Flickering Atari was previously used as the test environment in DRQN (Hausknecht & Stone, 2015), ADRQN (Zhu et al., 2018) and DVRL (Igl et al., 2018). The ViZDoom platform is a 3D FPS game for AI research. The agent in ViZDoom needs to navigate in the 3D environment to accomplish various tasks, such as gathering resources, shooting enemies and looking for the exit. Because the agent can only see in just one direction each time step, this naturally introduces partial observability to the task.

### 5.1 EXPERIMENT SETUP

We used some recent algorithms as baselines. Because the true state and transition function are unknown, only the methods which can be trained under model-free setting will be used for comparison. Baselines are listed as below:
**Deep Q-Networks (DQN) (Mnih et al., 2015):** A method which uses standard Bellman backup and uses the temporal difference error as the objective function.
**Deep Soft Q-Networks (DSQN) (Levine, 2018):** A method which uses soft Bellman backup and uses the temporal difference error as the objective function.
**Deep Recurrent Q-Networks (DRQN) (Hausknecht & Stone, 2015):** A method which applies LSTM on the top of the DQN.
**Action-Specific Deep Recurrent Q-Networks (ADRQN) (Zhu et al., 2018):** A method which extends the DRQN, i.e., it uses both the observation and action as inputs.

**Deep Variational Reinforcement Learning (DVRL)** (Igl et al., 2018) A method which combines sequential Monte Carlo and A2C (Dhariwal et al., 2017) to solve POMDPs.

## 5.2 Evaluation on Flickering Atari

Atari environments (Bellemare et al., 2013) are widely used as the benchmark for deep reinforcement learning algorithms due to its high dimensional observation spaces and numerous challenging tasks. Flickering Atari was introduced by (Hausknecht & Stone, 2015). In each running step, the observation may be obscured with a certain probability, i.e., the raw screen will be either fully observable or fully obscured with black pixels. The settings of experiments keep in line with DVRL (Igl et al., 2018), i.e., only one frame is used, the frameskip is set to four and each frame is obscured with a probability of 0.5.

We choose four Atrai games (i.e., Pong, ChopperCommand, DoubleDunk and Asteroids) for evaluation. Fig. 3 shows the screen shots of these games. When only given one frame as the input, these tasks are POMDPs because we cannot obtain the velocity of the moving object from a single observation. These tasks have high dimensional and continuous observation spaces, while discrete action spaces. The basic network architecture and hyper-parameters are similar to DQN (Mnih et al., 2015). For the recurrent neural networks, we use a sequence length of 5 for training. All the algorithms train for 10000,000 steps and run for 100 episodes during evaluation. The training details can be found in Appendix E.

Table 1 shows the performance results of different algorithms on these Atari games. We can see that our algorithms significantly outperform other baselines on three of the games and get close score to DVRL on ChopperCommand. Compared with DRQN and ADRQN, our method introduces inductive bias to the network, which helps state estimate and RL planning for POMDPs. Compared with DVRL, our method can achieve competitive performance with lower sampling complexity.

|  | DQN | DSQN | DRQN | ADRQN | DVRL | SVQN(GRU) | SVQN(LSTM) |
|---|---|---|---|---|---|---|---|
| Pong | -4.9($\pm$2.6) | -2.1($\pm$2.3) | 1.6($\pm$7.8) | 7($\pm$4.6) | 18.2($\pm$2.7) | **19.2**($\pm$1.3) | 18.6($\pm$1.9) |
| Chopper | 1350($\pm$731) | 1250($\pm$522) | 1090($\pm$409) | 1608($\pm$707) | **6602**($\pm$449) | 6005($\pm$258) | 5805($\pm$312) |
| DDunk | -16.2($\pm$3.1) | -16.9($\pm$2.8) | -14.4($\pm$3.2) | -12.9($\pm$3.6) | -6.0($\pm$1.3) | -5.8 ($\pm$1.4) | **-5.5**($\pm$1.2) |
| Asteroids | 935($\pm$410) | 940($\pm$320) | 871($\pm$340) | 1040($\pm$432) | 1539($\pm$73) | **1645**($\pm$102) | 1585($\pm$86) |

Table 1: Evaluation results of different models on flickering Atari. The values are the final evaluation scores after training for different algorithms. Values in parentheses indicate the standard deviation. Evaluations use Mann-Whitney rank test and bold numbers indicate statistical significance at the 5% level. Our algorithms outperform other baselines on three of the games and get close score to DVRL on ChopperCommand.

## 5.3 Evaluation on ViZDoom Tasks

We designed three tasks in ViZDoom as the evaluation tasks for our algorithm.

**Health Gather:** The agent needs to gather health supplies in a flat map. The agent will lose health every time step. Accordingly, if it can't gather enough health supplies, it will die and the game is over. At each time step, the agent will receive a reward of 0.001, which encourages the agent to live a longer life. When it collects one health supply, it will receive a reward of 1. The observation for the agent is the grayscale image of its vision and its health value. The agent has three actions to choose, i.e., **TURN LEFT**, **TURN RIGHT** and **MOVE FORWARD**.

**Health Gather v2:** This is a more difficult task than previous task with a more complex map. There is some poison in the map. When the agent encounters poison, it will lose health. We use the same reward scenario, observations and actions in **Health Gather**.

**Defend Center:** In this task, the agent stands in the center of a flat map. Monsters will walk toward the agent from the edge of the map. When the monsters touch the agent, the agent will lose health. The agent holds a gun with limited ammo, and it can kill the monsters by pressing down the shooting button. The agent gets reward of 4 by killing monsters, reward of -0.2 by using ammo and reward of

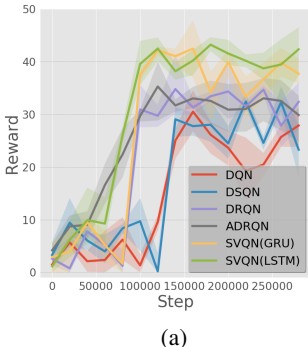 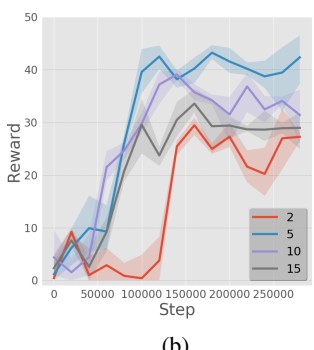 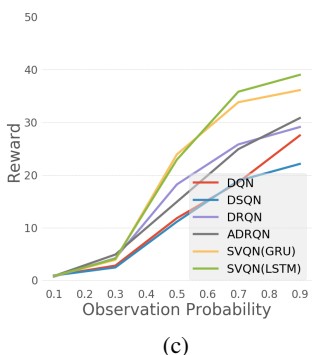

|     |     |     |
| (a) | (b) | (c) |

Figure 4: (a) Training curves on Health Gather. Both SVQN models outperform other baselines. (b) Performances of the models with different training sequence lengths on Health Gather. When the training sequence length is too long, it is hard for the algorithm to gather useful information through a long gradient flow. (c) Evaluation results of different models under different observation probabilities of the Health Gather, and all models are trained with full observation. The results show that SVQN models are more robust to the disturbance of the observation than other algorithms.

|  | DQN | DSQN | DRQN | ADRQN | SVQN(GRU) | SVQN(LSTM) |
|---|---|---|---|---|---|---|
| Health Gather | 27.8($\pm$1.4) | 23.3($\pm$1.7) | 32.4($\pm$2.0) | 29.9($\pm$1.6) | 37.7($\pm$0.9) | **42.3**($\pm$1.7) |
| Health Gather v2 | 9.8($\pm$1.5) | 6.7($\pm$2.1) | 13.2($\pm$2.5) | 14.3($\pm$1.2) | 15.8($\pm$1.1) | **18.2**($\pm$0.9) |
| Defend Center | 35.4($\pm$1.3) | 38.5($\pm$1.5) | 46.0($\pm$0.5) | 45.8($\pm$2.2) | **50.3**($\pm$2.1) | 48($\pm$1.4) |

Table 2: Evaluation results of different models on ViZDoom. The values are the final evaluation scores after training for different algorithms. Values in parentheses indicate the standard deviation. Evaluations use Mann-Whitney rank test and bold numbers indicate statistical significance at the 5% level. The SVQN models achieve the best performance on these three tasks.

-1 when losing health. The observation for the agent is the grayscale image of its vision, its health value and the ammo remained. The agent has three actions to choose, i.e., **TURN LEFT**, **TURN RIGHT** and **ATTACK**.

Fig. 3 shows the screen shots of these three tasks. All the algorithms are trained with the same basic network architecture and use the same hyperparameters. All the models take only one observation as input at each time step and the vision inputs are resized to the resolution of $84 \times 84$. For the recurrent neural networks, we use a sequence length of 5 for training. All the algorithms train for 300,000 steps and run for 20 episodes during evaluation. The discount factor $\gamma$ is set to 0.95, learning rate is 0.0001 and the Adam optimizer (Kingma & Ba, 2014) is used for training. The network architecture for these tasks and training details are shown in Appendix D.

Fig. 4(a) shows the training curves of different methods on the Health Gather. Both SVQN models outperform other baselines. Table 2 reports the final scores of different models on these three tasks. The SVQN models achieve the best performance on these three tasks. Experiments on ViZDoom indicate that generative models of SVQNs can improve agent's exploration ability in complicated unknown environment

## 5.4 ABLATION STUDY

**Training sequence length:** We study how the training sequence length $H$ impacts the algorithm's performance. We use the SVQN(LSTM) model and the environment of Health Gather for this study. Fig. 4(b) shows training processes of models with different training sequence lengths($H = 2, 5, 10, 15$). We can see that, when the sequence length is too short($H = 2$), the model can't gather enough historical information, which causes performance degradation. When the sequence length is too long($H = 15$), it is arduous for the network to do optimization, which will also

lead to performance degradation. This experiment also shows that with a limited training length, the algorithm can generalize to the time of arbitrary lengths during testing. Thanks to this generalization ability, we can train the agent with a fixed length of data instead of the full-length data, which can reduce the computation complexity.

**Observation Probability:** We also study whether different models are robust to the noise of the observation. We add a modification to the game Health Gather, i.e., at each time step, the observation of the screen is either fully revealed or fully obscured with a fixed observation probability $p$. Fig. 4(c) show the evaluation results of different models under different observation probabilities($p = 0.1, 0.2, 0.3, 0.4, 0.5, 0.6, 0.7, 0.8, 0.9$). All the models are trained under the standard environment with full observations. The results show that SVQN models are relatively robust to the disturbance of the observation compared to other algorithms.

## 6 CONCLUSIONS

We propose a novel algorithm named Sequential Variational Soft Q-Learning Networks (SVQN) to solve POMDPs with the discrete action space. SVQN is model-free and does not need to know the true state's representation. We apply generative models to deal with the conditional prior of hidden states and use a recurrent neural network to reduce the computational complexity, i.e., with a small length of training data, it can generalize to the test data with an arbitrary length. Our designed deep neural network can be trained end-to-end, which optimizes the planning and inference of hidden states jointly. Experimental results show that SVQN outperforms previous methods on challenging tasks and has the robustness to the disturbance of the observation. SVQN is also flexible and can be integrated with other maximum entropy reinforcement learning algorithms, such as soft actor-critic (Haarnoja et al., 2018). In the future, we will try to develop algorithms for POMDPs problems with the continuous action space.

### ACKNOWLEDGMENTS

This work was supported by the National Key Research and Development Program of China (No. 2017YFA0700904), NSFC Projects (Nos. 61620106010, U19B2034, U1811461), Beijing NSF Project (No. L172037), Beijing Academy of Artificial Intelligence (BAAI), Tsinghua-Huawei Joint Research Program, a grant from Tsinghua Institute for Guo Qiang, Tiangong Institute for Intelligent Computing, the JP Morgan Faculty Research Program and the NVIDIA NVAIL Program with GPU/DGX Acceleration.

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

## A  DERIVATION OF THE VARIATIONAL LOWER BOUND

$$\log p(\mathcal{O}_{0:T}, a_{0:T}, o_{1:T})$$

$$= \log \mathbb{E}_{q(s_{1:T}|\mathcal{O}_{1:T}, a_{0:T}, o_{1:T})} \left[ \frac{p(s_{1:T}, \mathcal{O}_{0:T}, a_{0:T}, o_{1:T})}{q(s_{1:T}|\mathcal{O}_{0:T}, a_{0:T}, o_{1:T})} \right]$$

$$\geq \mathbb{E}_{q(s_{1:T}|\mathcal{O}_{1:T}, a_{0:T}, o_{1:T})} \log \left[ \frac{p(s_{1:T}, \mathcal{O}_{0:T}, a_{0:T}, o_{1:T})}{q(s_{1:T}|\mathcal{O}_{0:T}, a_{0:T}, o_{1:T})} \right]$$

$$= \int q(s_{1:T}|\mathcal{O}_{1:T}, a_{0:T}, o_{1:T}) \log \left[ \frac{p(s_{1:T}, \mathcal{O}_{0:T}, a_{0:T}, o_{1:T})}{q(s_{1:T}|\mathcal{O}_{0:T}, a_{0:T}, o_{1:T})} \right] ds_{1:T}$$

$$= \int \sum_{t=1}^{T} q(s_{1:T}|\mathcal{O}_{1:T}, a_{0:T}, o_{1:T}) \log \left[ \frac{p(a_t)p(\mathcal{O}_t|s_t, a_t)p(s_t|s_{t-1}, a_{t-1})p(o_t|s_t)}{q(s_t|s_{t-1}, a_{t-1}, o_t)} \right] ds_{1:T}$$

$$= \sum_{t=1}^{T} \int q(s_{1:t}|\mathcal{O}_{1:t}, a_{0:t}, o_{1:t}) \log \left[ \frac{p(a_t)p(\mathcal{O}_t|s_t, a_t)p(s_t|s_{t-1}, a_{t-1})p(o_t|s_t)}{q(s_t|s_{t-1}, a_{t-1}, o_t)} \right] ds_{1:t}$$

$$= \sum_{t=1}^{T} \Big\{ \int q(s_{1:t}|\mathcal{O}_{1:t}, a_{0:t}, o_{1:t}) \log \left[ p(a_t)p(\mathcal{O}_t|s_t, a_t)p(o_t|s_t) \right] ds_{1:t}$$

$$+ \int q(s_{1:t}|\mathcal{O}_{1:t}, a_{0:t}, o_{1:t}) \log \left[ \frac{p(s_t|s_{t-1}, a_{t-1})}{q(s_t|s_{t-1}, a_{t-1}, o_t)} \right] ds_{1:t} \Big\} \qquad (19)$$

$$= \sum_{t=1}^{T} \Big\{ \int q(s_{1:t}|\mathcal{O}_{1:t}, a_{0:t}, o_{1:t}) \log \left[ p(a_t)p(\mathcal{O}_t|s_t, a_t)p(o_t|s_t) \right] ds_{1:t}$$

$$- \int q(s_{1:t-1}|\mathcal{O}_{1:t-1}, a_{0:t-1}, o_{1:t-1}) \mathcal{D}_{KL} \left[ q(s_t|s_{t-1}, a_{t-1}, o_t) || p(s_t|s_{t-1}, a_{t-1}) \right] ds_{1:t} \Big\}$$

$$= \mathbb{E}_{q(s_{1:T}|\mathcal{O}_{1:T}, a_{0:T}, o_{1:T})} \sum_{t=1}^{T} \Big\{ \log \left[ p(a_t)p(\mathcal{O}_t|s_t, a_t)p(o_t|s_t) \right]$$

$$- \mathcal{D}_{KL} \left[ q(s_t|s_{t-1}, a_{t-1}, o_t) || p(s_t|s_{t-1}, a_{t-1}) \right] \Big\}$$

$$\simeq \sum_{t=1}^{T} \Big\{ r(s_t, a_t) + \log \left[ p(a_t)p(o_t|s_t) \right]$$

$$- \mathcal{D}_{KL} \left[ q(s_t|s_{t-1}, a_{t-1}, o_t) || p(s_t|s_{t-1}, a_{t-1}) \right] \Big\}, \text{where } s_{1:T} \sim q(s_{1:T}|\mathcal{O}_{1:T}, a_{0:T}, o_{1:T})$$

## B  VARIATIONAL AUTOENCODERS

Variational auto-encoders (VAEs) (Kingma & Welling, 2013) are effective generative models that can recover complex distributions over the data space. In the VAE, there are observed data $x$ and the underlying causal factor $z$. Usually, it's intractable to do inference for the posterior $p(z|x)$. The VAE uses a variational function $q(z|x)$ to approximate the true posterior. The lower bound of the marginal distribution of the observed data is given by:

$$\log p(x) = \log \mathbb{E}_{q(z|x)}\left[\frac{p(x,z)}{q(z|x)}\right] \geq \mathbb{E}_{q(z|x)}\left[\log \frac{p(x,z)}{q(z|x)}\right], \tag{20}$$

and the lower bound can be equivalently written as :

$$\mathcal{L}(x) = -\mathcal{D}_{KL}(q(z|x)||p(z)) + \mathbb{E}_{q(z|x)}\left[\log p(x|z)\right], \tag{21}$$

where $\mathcal{D}_{KL}(\cdot||\cdot)$ is Kullback-Leibler divergence between two distributions. The approximate posterior $q(z|x)$ is often set as Gaussian $\mathcal{N}(\mu, \text{diag}(\sigma^2))$, where $\mu$ and $\sigma$ are the outputs of a non-linear function of $x$. The generative model $p(x|z)$ and inference model $q(z|x)$ can be trained jointly via standard backpropagation techniques.

## C  OPTIMIZE THE KLS IN EQ. (15)

We give out the final form of the loss function as below:

$$\begin{aligned}
\mathcal{D}_{KL}(p||q) &= \int p(x)\frac{p(x)}{q(x)}dx \\
&= \int p(x)\log p(x)dx - \int p(x)q(x)dx \\
&= \log\frac{\sigma_q}{\sigma_p} + \frac{\sigma_p^2 + (\mu_p - \mu_q)^2}{2\sigma_q^2} - \frac{1}{2} \\
&= \frac{1}{2}\left[2\log\frac{\sigma_q}{\sigma_p} + \frac{\sigma_p^2 + (\mu_p - \mu_q)^2}{\sigma_q^2} - 1\right] \\
&= \frac{1}{2}\left[-2\log\frac{\sigma_p}{\sigma_q} + \frac{\sigma_p^2 + (\mu_p - \mu_q)^2}{\sigma_q^2} - 1\right] \\
&= -\frac{1}{2}\left[2(\log(\sigma_p) - \log(\sigma_q)) + 1 - \frac{\sigma_p^2 + (\mu_p - \mu_q)^2}{\sigma_q^2}\right]
\end{aligned} \tag{22}$$

, where $p$ and $q$ are two Gaussian distributions. $\sigma_p, \sigma_q, \mu_q$ and $\mu_p$ are their standard deviations and means respectively.

# D NETWORK ARCHITECTURE

## D.1 ATARI

The network architecture for Atari is shown as below:

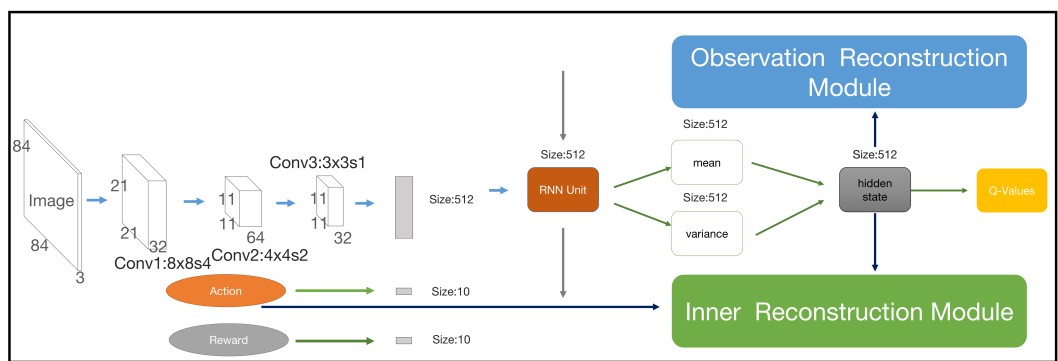

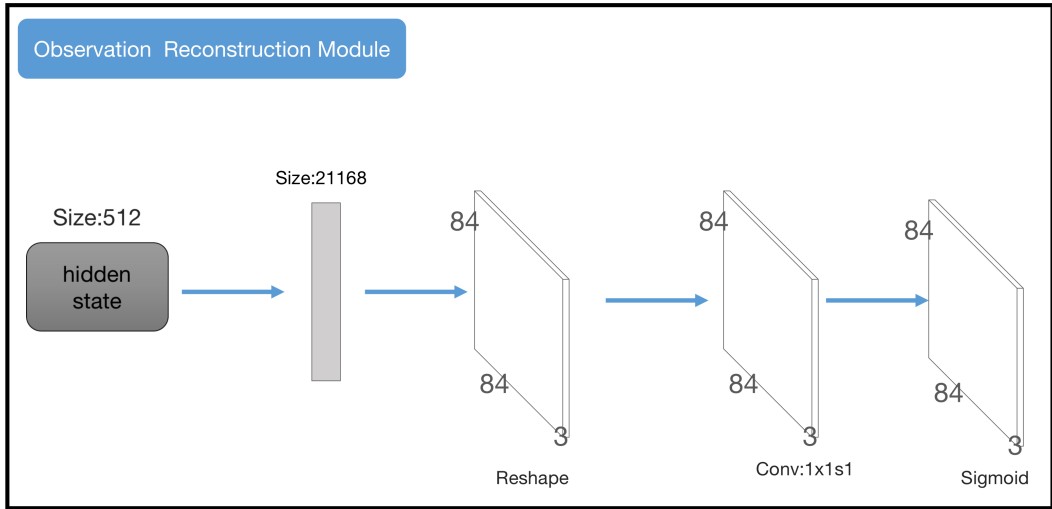

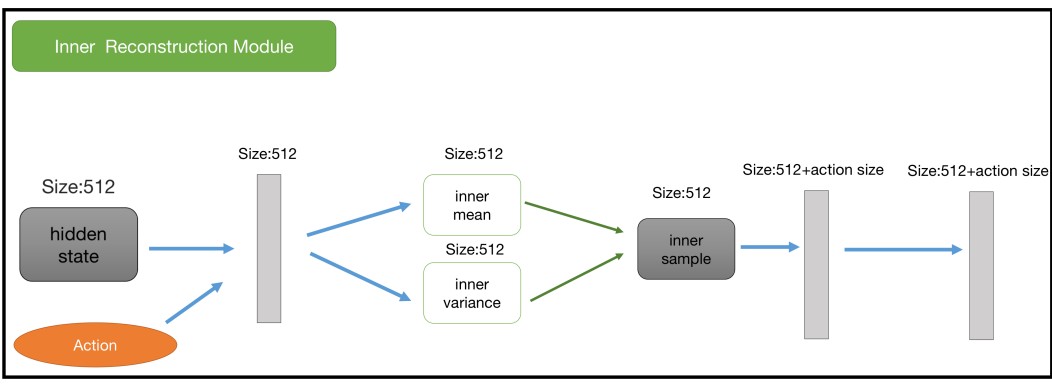

## D.2  VIZDOOM

The network architecture for ViZDoom is shown as below:

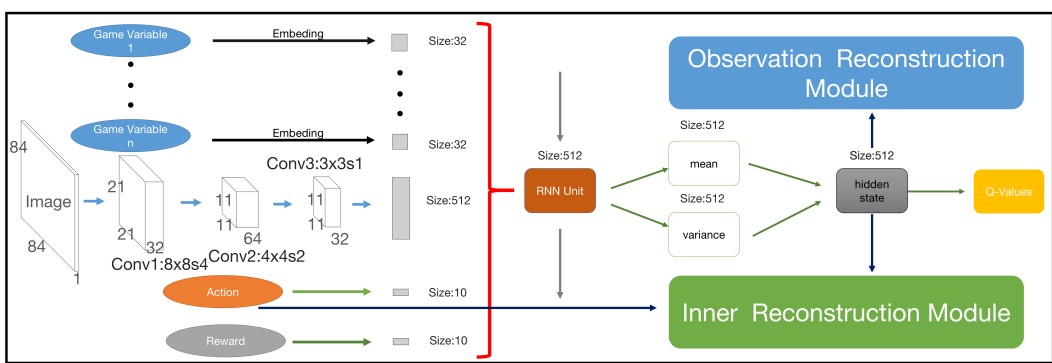

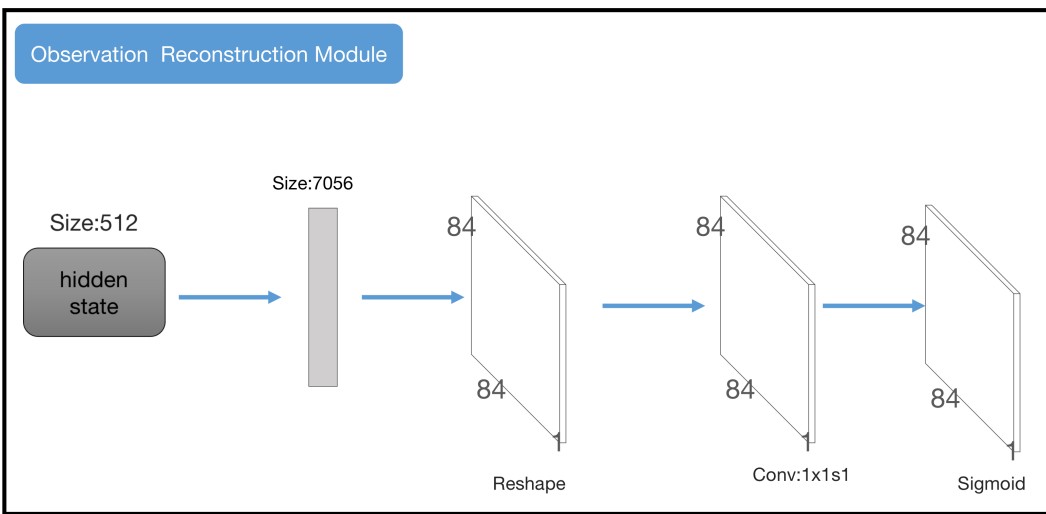

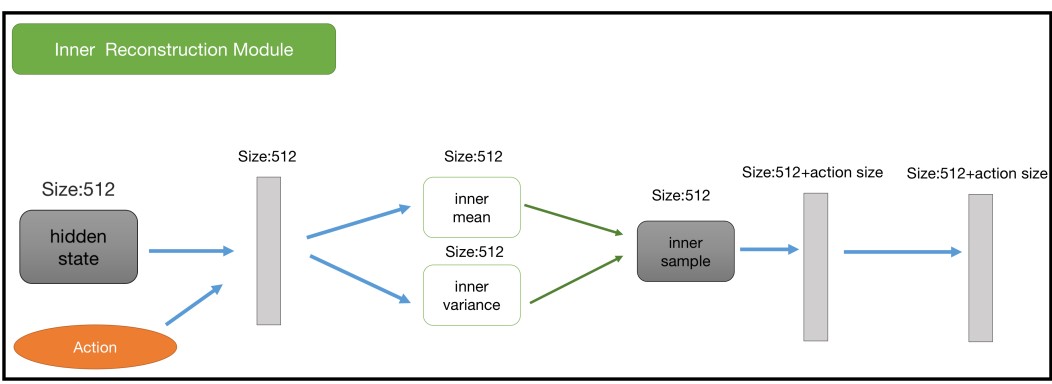

# E    TRAINING DETAILS

Training architechture:

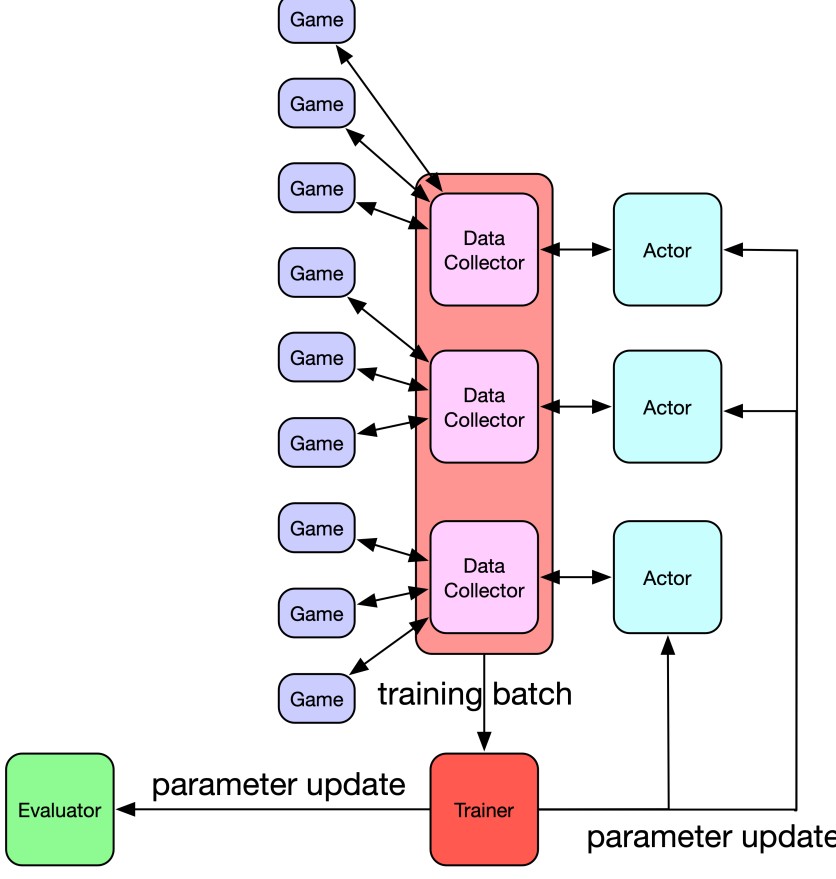

The data produce module, training module and evaluation module are run in parallel. The framework is implemented by Python and Tensorflow.

## E.1    ATARI

All the models take only one observation as input at each timestep and the vision inputs are resized to the resolution of $84 \times 84$. For the recurrent neural networks, we use sequence length of 5 for training. All the algorithms train for 1,0000,000 steps and run for 100 episodes during evaluation.

## E.2    VIZDOOM

All the algorithms are trained with the same basic network architectures and use the same hyperparameters. All the models take only one observation as input at each timestep and the vision inputs are resized to the resolution of $84 \times 84$. For the recurrent neural networks, we use sequence length of 5 for training. All the algorithms train for 300,000 steps and run for 20 episodes during evaluation. The discount factor $\gamma$ is set to $0.95$, learning rate is $0.0001$ and the Adam optimizer is used for training.

