# OpenReview forum: "SVQN: Sequential Variational Soft Q-Learning Networks"
_ICLR.cc/2020/Conference — Accept (Poster)_

### Official Review · AnonReviewer3 · 2019-10-17
**Official Blind Review #3**

**Rating:** 8

**Review:**

This paper proposes a new sequential model-free Q-learning methodology for POMDPs that relies on variational autoencoders to represent the hidden state. The approach is generic, well-motivated and has  clear applicability in the presence of partial observability. The idea is to create a joint model for optimizing the hidden-state inference and planning jointly. For that reason variational inference is used to optimize the ELBO objective in this particular setting. All this is combined with a recurrent architecture that makes the whole process feasible and efficient.

The work is novel and it comes with the theoretical derivation of a variational lower bound for POMDPs in general. This intuition is exploited to create a VAE based recurrent architecture. One motivation comes from maximal entropy reinforcement learning (MERL), but which has the ad hoc objective of maximizing the policy entropy. On the other hand SVQN optimizes both a variational approximation of the policy and that of the hidden state. Here the rest terms of the ELBO objective can be approximated generatively and some of them are conditioned on the previous state which calls for a recurrent architecture. The other parts are modeled by a VAE.

The paper also explores two different recurrent models in this context: GRU and LSTM are both evaluated. Besides the nice theoretical derivation the paper presents compelling evidence by comparing this approach to competing approaches on four games of the flickering ATARI benchmark and outperforming the baselines significantly. Also both the GRU and LSTM version outperforms the baseline methods on various tasks of the VIZDoom benchmark as well.

In general, I find that this well written paper presents a significant progress in modelling POMDPS in a model-free manner with nice theoretical justification and compelling empirical evidence.






**Experience Assessment:**

I have read many papers in this area.

**Review Assessment: Checking Correctness Of Derivations And Theory:**

I assessed the sensibility of the derivations and theory.

**Review Assessment: Checking Correctness Of Experiments:**

I assessed the sensibility of the experiments.

**Review Assessment: Thoroughness In Paper Reading:**

I read the paper at least twice and used my best judgement in assessing the paper.

---

> ### Author Response · Authors · 2019-11-11
> **Response to AnonReviewer #3**
>
> We thank Reviwer #3 for the valuable comments and the appreciation of our novel contributions.

---

### Official Review · AnonReviewer1 · 2019-10-22
**Official Blind Review #1**

**Rating:** 3

**Review:**

The paper proposes SVQN, an algorithm for POMDPs based on the soft Q-learning framework which uses recurrent neural networks to capture historical information for the latent state inference. In order to obtain this formulation, the author first derive the variational bound for POMDPs and then present a practical algorithm.

The key idea of the paper is to replace DQN with Soft Q-learning that already demonstrated better performance on a variety of tasks. This seems to be an obvious extension of DRQNs (Hausknecht & Stone, 2015) even though it did not appear in the literature.

The authors evaluate the final algorithm on a set of ALE and DoomViz tasks. The algorithm outperforms the previous methods, in particular, DRQNs. The set of tasks and prior methods is adequate.

Overall, the contribution of the paper is not significant enough to be accepted to ICLR.


**Experience Assessment:**

I have published one or two papers in this area.

**Review Assessment: Checking Correctness Of Derivations And Theory:**

I assessed the sensibility of the derivations and theory.

**Review Assessment: Checking Correctness Of Experiments:**

I assessed the sensibility of the experiments.

**Review Assessment: Thoroughness In Paper Reading:**

I read the paper at least twice and used my best judgement in assessing the paper.

---

> ### Author Response · Authors · 2019-11-11
> **Response to AnonReviewer #1**
>
> We thank the reviewer for the valuable comments. However, we humbly disagree on the primary concern that our paper is an "obvious extension" of DRQNs (Hausknecht & Stone, 2015).
>
> Firstly, it is not just an "obvious extension" of DRQNs. As stated in the third paragraph of Section 1, both DRQNs and its improved version of Action-specific Deep Recurrent QNetwork (ADRQN) (Zhu et al., 2018) fail to utilize the Markov property of the state in POMDPs, because they just represent the state as latent variables of neural networks. To solve this problem, our algorithm starts from the graphical model representation of POMDPs, which is very intuitive and generic, and then we derive the ELBO and finally lead to the design of the neural network. In the experiments, we show that our method outperforms both DRQN and ADRQN by a large margin on several challenging tasks. And our ablation study also shows that our method is more robust to the disturbance of the observation. Overall, as agreed by Reviewer #3, we provide a novel solution to POMDPs, which is better than DRQNs and ADRQN in terms of both theoretical formulation and empirical results.
>
> Secondly, it is not just "replace DQN with Soft Q-learning". The reason why we use soft Q-learning (i.e., Maximum Entropy Reinforcement Learning) has been explained in Section 3.2.  As we want to solve the POMDPs under a unified graphical model, we derive the ELBO of POMDPs and design generative models to handle the inference of the hidden states. Moreover, we apply additional approximate functions to tackle the conditional prior problem (as stated in Section 4.2). Finally, to train the generative models and the planing algorithm jointly, we design a recurrent neural network to reduce the computation complexity. We also explore two different recurrent models in our context: GRU and LSTM, and both of them outperform the baseline methods on various tasks. Fig. 4(a) also shows that our jointly training process is more effient than other baselines. Table 1 shows that our algorithm is more powerful than the naive soft q-learning algorithm (DSQN). To summarize, we have done much work to deal with the challenges in POMDPs and improve the performance of our algorithm.
>
> We hope our answers can address your concerns.

---

### Decision · Program_Chairs · 2019-12-19

**Decision:**

Accept (Poster)

**Comment:**

The paper proposes a novel model-free solution to POMDPs, which proposes a unified graphical model for hidden state inference and max entropy RL. The method is principled and provides good empirical results on a set of experiments that relatively comprehensive. I would have liked to see more POMDP tasks instead of Atari, but the results are good. Overall this is good work.